# Implementation and impact of a dental preventive intervention conducted within a health promotion program on health inequalities: A retrospective study

Hélène Pichot[1,2]☯*, Bruno Pereira[3]☯, Elodie Magnat[2]‡, Martine Hennequin[1,4]‡, Stéphanie Tubert-Jeannin🆔[1,4]☯*

1 CROC, Université Clermont Auvergne, Clermont-Ferrand, France, 2 Health and Social Agency of New Caledonia, Nouméa, New Caledonia, 3 CHU Clermont-Ferrand, Délégation à la recherche clinique et à l'innovation, Clermont-Ferrand, France, 4 CHU Clermont-Ferrand, Service d'Odontologie, Clermont-Ferrand, France

☯ These authors contributed equally to this work.
‡ These authors also contributed equally to this work.
* stephanie.tubert@uca.fr (STJ); helene.pichot@ass.nc (HP)

**Data Availability Statement:** The data underlying this study are available from Zenodo using DOI: 10.5281/zenodo.3670886.

## Abstract

### Background

The objective of this retrospective survey was to evaluate after one year, the conditions and impacts of a dental sealant intervention conducted in New Caledonia, within a health promotion program. A greater or at least equivalent quality and impact of the intervention was expected for children living in socially deprived regions with the greatest health needs.

### Methods

The study population was the schoolchildren, aged 6 years in 2016, who benefited from the dental sealant program (n = 2532). The study sample was randomly selected in 2017 from that population (n = 550). The children's dental status was evaluated at school in 2017 and compared with that recorded in 2016 during the sealant intervention allowing the calculation of the retention rates and one-year carious increment on first permanent molars. Socio-demographic variables (gender, public/private school) and conditions of sealant placement (school/dental office, presence of a dental assistant) were recorded. The carious increment was explained using a mixed multiple random-effects regression. A mediation analysis was conducted to assess the respective contributions of the retention rates and the region of origin on caries increment.

### Results

The participation rate was very high (89%) and on average, children had 83% of their dental sealants present after one year, 31% fully and 52% partially present. Caries increment varied depending on the sealant retention rate as well as on the region (North, South, Islands).

**Funding:** The author(s) received no specific funding for this work

**Competing interests:** The authors have declared that no competing interests exist.

**Abbreviations:** NC, New Caledonia; OHP, Oral Health Promotion program; ICDAS, International Caries Detection and Assessment System; $ICDAS_{4-6}$, Untreated carious lesions recorded when dentine is involved; $DT_{1st\ molars}$, Number of untreated carious lesions on first permanent molars; $\Delta DT_{1st\ molars}$, One-year caries increment for first permanent molars; CNIL, Comité National Informatique et Liberté (French ethical committee for data management).

The mediation analysis showed that living in a deprived area (The Islands) was a strong determinant for high caries increment particularly when the retention rates were low.

## Conclusions

This study showed a high participation rate and acceptable effectiveness as measured with the one-year retention rates, for a fissure sealant intervention conducted in real-life conditions and integrated in a large health promotion program. Nevertheless, the intervention was not effective enough to totally balance the influence of health determinants, especially in socially deprived sectors characterized by greater dental needs.

## Introduction

Oral diseases remain a major health problem in both developing and developed countries [1]. Indeed, the prevalence of untreated oral conditions is high worldwide. Low-income and socially- or medically-disadvantaged populations experience higher rates of chronic diseases and this gradient is particularly apparent for oral diseases [2].

Since oral health inequalities come across the whole social body according to a gradient, the challenge is to take into account that gradient when evaluating public health interventions. An efficient intervention might have no effect on health inequalities if all socioeconomic groups benefit equally. And it may even increase health inequalities if the wealthiest groups benefit more [3]. Indeed, the more affluent communities are often in a better position to implement health promotion programs not only for financial reasons but also because of environmental factors or manpower resources. Unfortunately, there is very little evidence concerning the equity effects of health-related interventions. Thus, studies assessing the impact of universal interventions on children with various social or environmental backgrounds are needed to identify the differential effects as well as the impact of the implementation conditions [4].

New Caledonia (NC) (245,580 inhabitants) is a French Oceanian overseas territory, with extensive administrative autonomy. The population is a mix of 40% Kanak (indigenous population of New Caledonia), 30% White European (Caledonians and Metropolitans), 10% Polynesian (Wallisians) and South-East Asian or Vanuatu people. The population varies depending on the region, with more Kanak in the North and the Islands and more Polynesians, white Europeans and Asians in the South. Three quarters of the population live in the South, where the economic activity is concentrated, and 39% of the population lives in Nouméa city. New Caledonia is one of the richest countries in the South Pacific area but there are strong social and economic disparities between regions (North, Islands, South) [5]. As an example, the proportion of households having a computer at home varied in 2014 from 37% in the Islands, to 53% in the North and 76% in the South.

A study conducted in 2011–2012 has evaluated the dental and weight status of 6-, 9- and 12-yr-old children in NC. The prevalence of untreated dental caries was almost 60% among 6- and 9-yr-olds and about 50% among 12-yr-olds. Caries experience was unevenly distributed in the population, with one third of 12-yr-olds having more than five untreated carious teeth. The number of carious lesions was related to their unfavourable lifestyle, deprived social status and lack of preventive dental care [6]. The prevalence of overweight and obesity was high and greatly increasing between the ages of 6 (respectively 10.8% and 7.8%) and 12 (respectively 22.2% and 20.5%), with one third of the 12-yr-olds having an excess of abdominal adiposity. Geographical location (region), ethnicity, tooth-brushing frequency and masticatory function were significant risk factors for oral diseases and overweight/obesity [7]. People with the lowest

levels of resources also showed difficulties attending for dental care despite the public medical/dental coverage existing in NC.

That study highlighted the need for new strategies aimed at improving children's oral and general health and at reducing inequalities. An alternative approach to traditional healthcare was thus chosen in collaboration with NC local populations and government, based on the principles of Oral Health Promotion (OHP) [8] with the development of—1—health education in schools in order to promote the adoption of health promoting behaviours [9], 2—preventive interventions, such as the implementation of tooth brushing at school, for the child population, 3—a reorientation of dental services towards more effective and preventive interventions. The OHP program ("Mes dents, Ma santé") was developed in connection with other health programs related to the prevention of other chronic diseases such as rheumatic heart disease or obesity. The common risk factor approach was thus privileged in order to address risk factors common to those chronic conditions within the context of the local environment [10].

Within this framework, the dental sealant program, which has existed since 2009, was renewed in order to ensure its quality and that all 6-yr-old schoolchildren can benefit from it. Indeed, dental sealants have been shown to be an effective preventive intervention when evaluated in real-life school conditions [11,12]. The program is now coordinated by the NC health agency in connection with regional educational and health authorities, health-fund bodies and the dental profession. A training course and a standardized protocol were implemented in 2014 in order to ensure the quality and relevance of sealant applications by dentists, in schools or public dental services.

The objective of this retrospective study was to evaluate after one year, the dental sealant program conducted in 2016 among 6-yr-old children in NC, by measuring the participation and retention rates [13]. The aim was also to evaluate the influence of the quality of the intervention (retention rates) on the one-year caries increment, while taking into account initial dental status, conditions of sealant placement and social determinants such as the region of origin. The objectives of this study are positioned within the general aim of taking into account oral health inequalities when assessing public health interventions. Indeed, to ensure that public health interventions do not increase health inequalities, a greater or at least equivalent quality and health outcome of those interventions are expected in socially deprived sectors with greatest health needs compared to more favoured areas [3].

## Population and methods

### Population and study sample

The study population was the group of schoolchildren aged 6 in 2016 who benefited from the dental sealant program; the program concerned 4329 children, was effectively offered to 3774 (87%) children (351 in the Islands, 908 in the North and 2515 in the South); 2532 children participated (89% in NC, 92% in the Islands, 88% in the North and 89% in the South).

The number of children to be selected in the 2017 study from that population was calculated (n = 500) to ensure the precision of the retention rates' estimates (rate = 50%, precision 5%) and to highlight an effect size above 0.5 when considering variations between regions. This number was also calculated following the criteria used in multidimensional and mediation analyses [14–16]. Children were randomly selected using a computerised, clustered sampling method with a probability proportional to the regions' population.

### Ethics approval and consent to participate

Study approvals were obtained from the NC educational and health institutions. Schools were approached through local educational authorities. Data were recorded in a file registered

within the French ethical committee for data management (CNIL -Commission Nationale Informatique et Liberté, No. kpP1390145R). Information letters and consent forms were sent to the parents. The children whose parents returned written consent were examined. In order to prevent negative impact on the participation rate in the OHP program, it was decided not to collect individual social information or behavioural data at this stage.

## Fissure sealant program

The age of 6 was chosen because school is compulsory from this age and it allowed the sealant to be applied on first permanent molars as soon as they erupted. The recommendation for the dentists is to seal all non-carious or with only very early enamel lesions (ICDAS$_1$) erupted first permanent molars whatever the caries risk [17]. Caries risk is very high in NC and it was considered not feasible nor pertinent to differentiate children according to caries risk. Photopolymerised resin sealants (3M Cleanpro) were applied with cotton rolls for saliva isolation [18,19]. Sealants were applied either at school in a mobile dental surgery or in public dental offices. Thirty general dental practitioners performed the 2016 dental examinations and applied the fissure sealants. They were recruited by the NC health agency within public regional health services and through the council of private dentists. The practitioners were assigned to one or several schools, depending on feasibility criteria that were mostly dependent on geographical locations. Children from the same school were examined by the same practitioner. Prior to the start of the program, all the practitioners underwent a training course consisting of a presentation of the NC OHP program, the sealant application protocol, illustrations with clinical situations and training about the data collection process.

## Study variables

The children's dental status was evaluated in 2017 at school. Radiographs were not used. Six general dental practitioners performed the children's examinations, after having been calibrated through a training course described in a previous study [6]. The presence of dental sealants was recorded. A sealant was considered as fully or partially present using standardised criteria [19]. For each child, the number of teeth with partially remaining sealant and fully-intact sealant was recorded. The clinical information collected in 2017 was connected to the 2016 individual data. The number of sealants applied in 2016 was compared with the number of sealants present during the 2017 examination. This allowed the calculation of three retention rates for each child; the complete retention rate (based on fully-present sealants), the partial retention rate (partially present) and the global retention rate (fully or partially present).

Dental caries were diagnosed at the dentinal threshold level (ICDAS$_{4-6}$) for deciduous and permanent teeth [20]. The dental status (DT $_{first\ molars}$: number of carious first permanent molars) recorded in 2016 during the sealant intervention was compared with that of the 2017 dental examination. This allowed the calculation of the one-year carious increment: $\Delta DT_{1st\ molars} = DT_{first\ molars}$ in 2017—DT $_{first\ molars}$ in 2016.

Socio-demographic variables (gender, region, public or private school) were recorded from the school registers. The setting where dental sealants were applied in 2016 (school vs dental office) and the presence of a dental assistant during the sealing procedure were also retrieved for each child.

## Data analysis

Data errors were corrected before analysis. The statistical analyses were performed using the Stata statistical software package, version 13.

Retention rates were expressed with 95% confidence intervals (CI). The influence of initial dental status, conditions of sealant placement and the child's profile (gender, school, region) was explored using bivariate analysis.

Variations in the distribution of the study variables (retention rates, initial and 2017 dental status, condition of sealant placement, sociodemographic profile, caries increment) depending on the region (an ecological variable representing a major social determinant in NC) were also explored using bivariate analysis. Chi-squared, ANOVA or Student t-tests were used depending on the variables being considered. For the main variables, when p-value was significant, a Tukey-Kramer post-hoc test was performed to take into account multiple comparisons.

A mixed multiple random-effects regression was used to evaluate the influence of the quality of the intervention (retention rates), initial dental status, conditions of sealant placement and the region of origin on the one-year caries increment. Mixed models were used including the examiner parameter as random effect [21–24]. The caries increment was not normally distributed (even with a normality transformation), and thus the 0-inflated Poisson model was used. The interactions between factors were tested.

We conducted a mediation analysis to assess the respective contributions of the retention rate and the region on caries increment. A mediation proportion was estimated, indicating how much of the whole increment value provided by an independent variable (the region) can be explained by the indirect path in which changes in this independent variable drives a change in the mediator (retention rate), and changes in the mediator then affect outcome [25,26]. We performed multilevel mediation analysis with other explanatory variables and examiner effect being integrated. Three models were conducted depending on the region being considered as the reference (A—South, B–Islands, C–North). The global and complete retention rates were considered separately. Results were summarized using a graph giving the mediation proportion and significance of the mediation analysis associations.

## Results

Of the children selected in the 2017 sample, 459 children participated and were examined. Among the participants, 48% were girls, 84% attended public schools, 9% lived in the Islands, 29% in the North and 62% in the South.

On average, children still had 83% (95%CI: 79.9%-86.2%) of their dental sealants after one year; 31.1% (27.8%-34.5%) were still fully present and 51.9% (48.6%-55.1%) were partially present. Retention rates did not vary depending on the children gender, type of school (private vs public) or the conditions of sealant placement (dental office vs school, presence of a dental assistant). Dental status in 2016, as well as the number of sealants applied in 2016, were not associated with retention rates. Retention rates appeared to be slightly better in the North and lower in the South (Table 1 and Fig 1).

In 2016, the mean number of sealed teeth was 3.34 (1.01) with 39 children (8%) having had one tooth sealed and 295 (64%) four teeth sealed. It was found that 31% of the children benefited from the sealant program within schools and a dental assistant was present in 84.9% of the cases.

The South showed a very high rate of sealant application at school with an assistant. In 2016, caries prevalence was high with non-treated carious lesions in eight children out of ten in the Islands (Table 1). In 2017, all children had their 4 first permanent molars erupted with no filling being recorded. It was observed that 63.6% of the children (58.6%-67.7%) had untreated carious lesions on permanent or temporary teeth. The 2017 caries prevalence varied depending on the region. In total, 168 first molars had carious lesions with a mean number of

**Table 1. Description of the study variables, for the whole sample and per region.**

| | Total n = 459 | North n = 132 | Islands n = 43 | South n = 284 | P |
|---|---|---|---|---|---|
| **RETENTION RATES** | | | | | |
| Partial retention rate | 0.52 (0.35) * | 0.59 (0.38) | 0.45 (0.32) | 0.50 (0.34) | 0.004! |
| Complete retention rate | 0.31 (0.36) | 0.30 (0.36) | 0.42 (0.37) | 0.30 (0.36) | NS |
| Global retention rate | 0.83 (0.35) | 0.89 (0.31) | 0.87 (0.45) | 0.80 (0.34) | 0.02! |
| **CONDITIONS: SEALANT APPLICATION** | | | | | |
| % of sealants placed at school | 69.1% | 44.7% | 25.6% | 87.0% | <0.001 |
| Presence of a dental assistant (n = 444) | 84.9% | 64.7% | 73.2% | 95.1% | <0.001 |
| **SOCIODEMOGRAPHIC PROFILE** | | | | | |
| % in private schools | 15.9% | 19.7% | 41.9% | 10.2% | P<0001 |
| % of girls | 48.0% | 40.2% | 55.8% | 50.4% | NS |
| **INITIAL DENTAL STATUS** | | | | | |
| Number of First molars present in 2016 | 3.63 (0.81) | 3.64 (0.76) | 3.47 (0.88) | 3.64 (0.82) | NS |
| % of children with caries in 2016$^£$ | 63.6% | 69.7% | 79.0% | 58.5% | 0.007 |
| Number of carious first molars in 2016 | 0.19 (0.54) | 0.20 (0.57) | 0.14 (0.41) | 0.19 (0.53) | NS |
| Number of sealants applied in 2016 | 3.34 (1.01) | 3.41 (0.92) | 3.21 (1.06) | 3.32 (1.03) | NS |
| **2017 DENTAL STATUS** | | | | | |
| Number of carious first molars in 2017 | 0.35 (0.84) | 0.32 (0.81) | 0.65 (0.90) | 0.37 (0.84) | 0.06 |
| Number of sealants remaining in 2017 | 2.82 (1.26) | 3.08 (1.23) | 2.74 (1.36) | 2.72 (1.25) | 0.03 |
| % of children with caries in 2017$^£$ | 63.0% | 68.9% | 88.4% | 56.4% | <0.001 |
| **ONE-YEAR CARIES INCREMENT** | | | | | |
| Caries increment: $\Delta DT_{1st\ molars}$ | 0.18 (0.76) | 0.12 (0.59) | 0.51 (0.83) | 0.15 (0.81) | <0.001§ |

n: number of subjects,

* mean (standard deviation) for the proportion of sealants remaining per child,

p: p value for Chi square test or ANOVA.

£ permanent and temporary teeth,

! the South region differed from the North region (Tukey-Cramer).

§ the Islands differed from the North and from the South (Tukey-Cramer).

decayed teeth per child of 0.35 (0.84). The mean number of sealants on first permanent molars was 2.82 (1.26) with only 25 children having no sealants and 43% with all their molars sealed.

Carious increment $\Delta DT_{1st\ molars}$ was 0.18 (0.76) and varied depending on the region with a much higher caries increment in the Islands (Table 1). The multidimensional model showed that caries increment varied significantly depending on the global sealant retention rate as well as the region (Table 2). Children living in the South and North and children with a high retention rate experienced less new carious lesions. The children's sociodemographic profile, modalities of sealant application as well as initial dental status were not significant factors. There was a significant examiner effect that was managed by using a mixed multiple random-effects regression. The same findings were nearly obtained when considering the complete retention rate (Regression Coefficient: -0.54, 95% CI: -0.74; -0.34, p<0.0001).

A multilevel mediation analysis was performed to evaluate if the retention rate was a mediator of the effect of the region on caries increment. The direct association between the region and caries increment remained significant only in Model B. In models A and C, the retention rate was linked to the region. The modalities of sealant application significantly influenced that relationship for the three models. For Model B, the influence of the region increased when retention rate was considered as a mediator. This means that living in the Islands was a strong determinant for high caries increment particularly when the retention rate was low. For

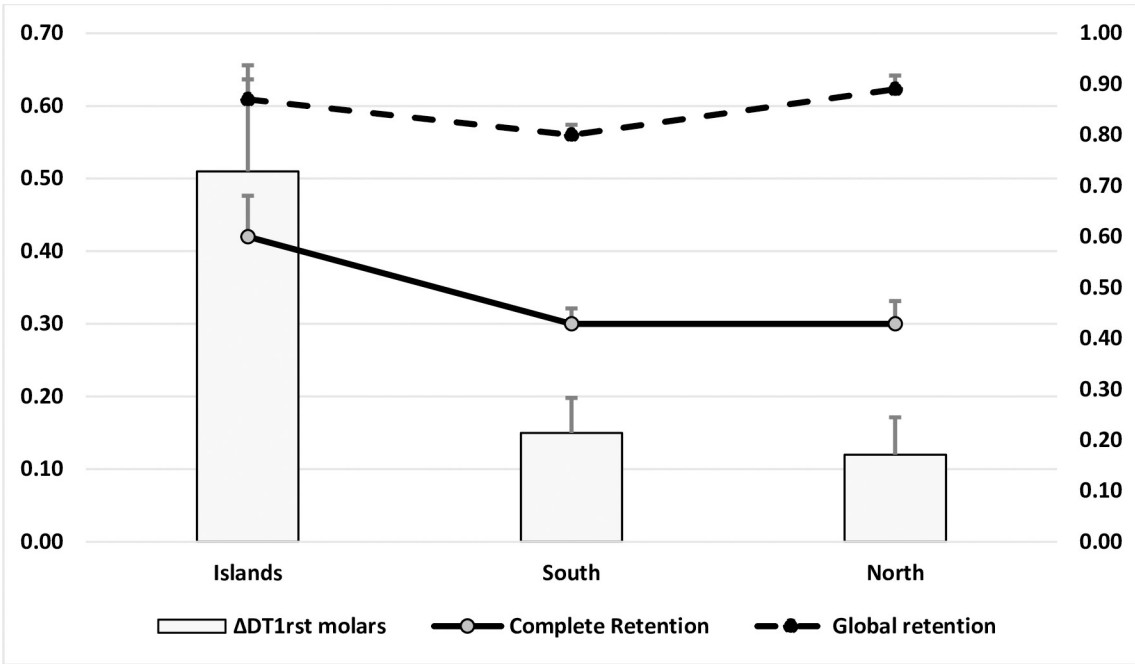

**Fig 1. Complete and global retention rates and caries increment ($\Delta DT_{1st\ molars}$) depending on the region and setting of sealant placement.**

Models A and C, the retention rate was not a mediator for the relationship between the region and caries increment. The mediator effect was lower for the complete retention rate (35%) as compared to the global retention rate (38%) (Fig 2).

## Discussion

The objective of this retrospective study was to evaluate the quality and the impact of the dental sealant program on oral health inequalities. Findings showed that the participation rate was very high and that on average, children had approximately 80% of their dental sealants present after one year, 30% being fully present and 50% partially present. The aim was also to evaluate

**Table 2. Significant regression coefficients derived from the mixed multiple random-effects regression with one-year carious increment ($\Delta DT_{1st\ molars}$) as dependent variable.**

|  | $\Delta DT_{1st\ molars}$ | | |
|---|---|---|---|
|  | **RC** | **95% CI** | ***p*** |
| **Region** |  |  |  |
| North | -0.30 | (-0.61; -0.01) | *0.05* |
| South | -0.38 | (-0.71; -0.05) | *0.02* |
| Islands = reference |  |  |  |
| **Global retention** | -0.28 | (-0.48; -0.08) | *0.005* |

RC: Regression coefficient, 95%CI: Confidence Interval, p: p value,

Explanatory variables integrated in the model: Child's gender, type of school (private vs public), region (Islands, North, South), dental status in 2016 (number of first molars and % with carious lesions), conditions of sealant placement (school vs dental office, with or without dental assistant), examiner effect, p = 0.002

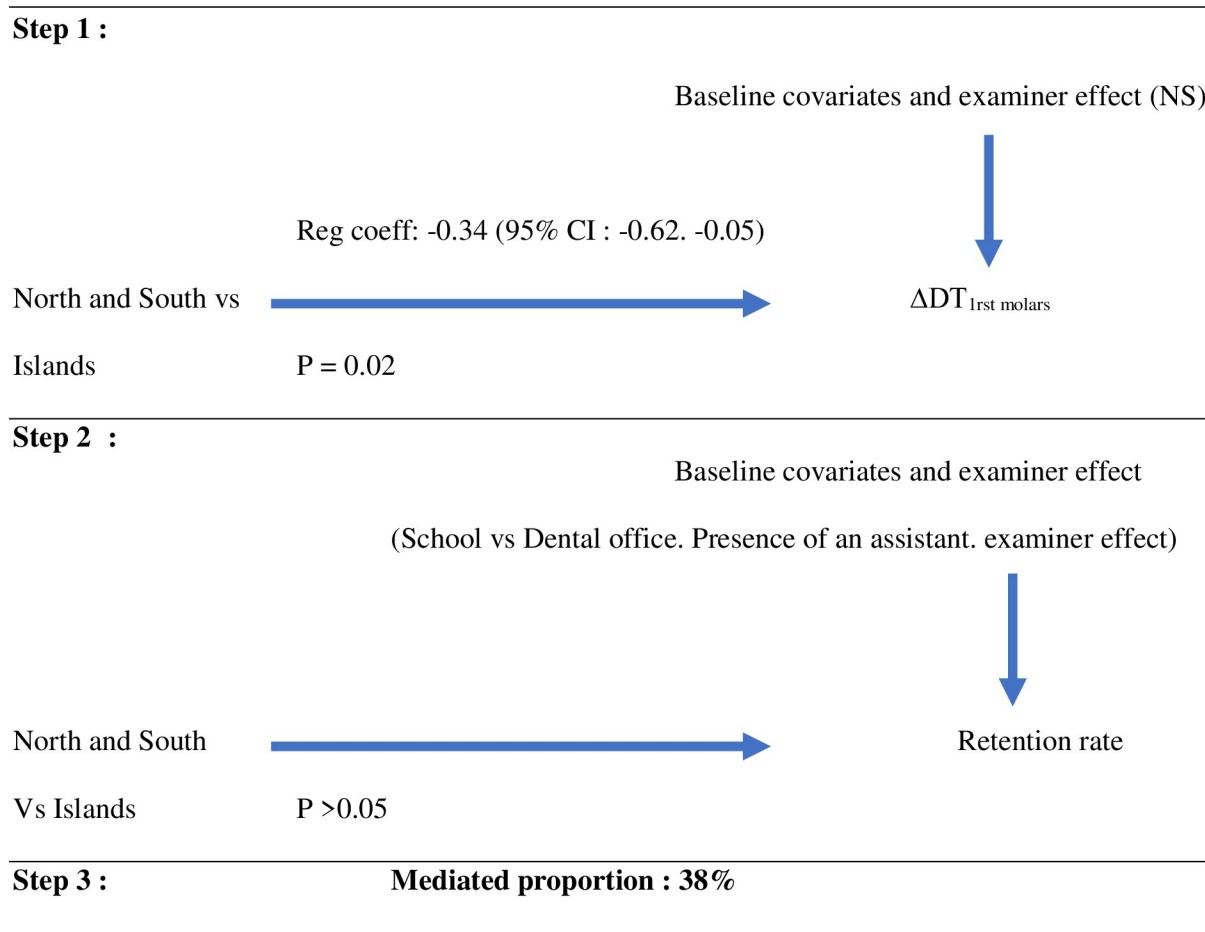

**Step 1 :**

Baseline covariates and examiner effect (NS)

Reg coeff: -0.34 (95% CI : -0.62. -0.05)

North and South vs Islands

$\Delta DT_{\text{1rst molars}}$

P = 0.02

**Step 2 :**

Baseline covariates and examiner effect

(School vs Dental office. Presence of an assistant. examiner effect)

North and South Vs Islands

Retention rate

P >0.05

**Step 3 :**  **Mediated proportion : 38%**

Baseline covariates and examiner effect (NS)

p<0.007

North and South vs Islands

Retention rate

p<0.001

$\Delta DT_{\text{1rst molars}}$

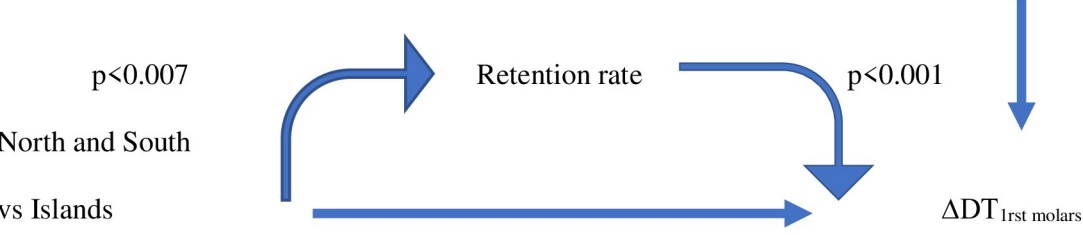

Retention rate : -0.33 (95% CI : -0.53. -0.13)

Region : -0.37 (95% CI: 0.65; 0.10)

**Fig 2. Mediation analysis of caries increment (Model B).** Tested mediator: Global retention rate. Independent variable: Region (Model B: the Islands compared to the North and South). Baseline risk co-variates: gender, type of school (private vs public), dental status in 2016 (number of first molars and % with carious lesions), conditions of sealant placement (school vs dental office, with or without dental assistant), examiner effect (as random effect). Step 1: The first step in our mediational analysis was the finding that belonging to the Islands region as compared to the North and South regions had a measurable impact on caries increment after accounting for baseline risk covariates. Step 2: Second, we checked if the retention rate (mediator) was related with the region, after accounting for baseline risk covariates. Step 3: Finally, a multilinear regression (mixed model) calculated the influence of the region on the tested mediator (retention rate). Subsequently, we jointly calculated the influence of the mediator and the direct effect of the independent variable on caries increment after accounting for baseline risk covariates. This last step shows that retention rate partially mediates [38%. P = for the average causal mediation effect (ACME)] the original effect of the region on caries increment and consequently remains directly associated with caries increment in an independent manner. The mediator (retention rate) and the main independent variable (Region) are assessed as binary variables.

the impact of retention rates, social determinants and other explanatory factors on caries increment for first permanent molars. Caries increment varied depending on the retention rates as well as on the region. The mediation analysis showed that living in a more deprived area was a strong determinant for high caries increment particularly when the retention rates were low.

The notion of retention is crucial because the main function of sealants is to form an efficient physical barrier between the enamel surface and the oral environment. Thus, retention rates are used as clinical evaluation criteria to measure effectiveness. The mean retention rate in this study is comparable to global retention rates found in other pragmatic studies that have evaluated the school sealant program [27–29]. However, the mean complete retention rate observed here (30%) was relatively low as compared to other studies with much higher rates (50–70%) [13,30–33]. The high proportion of partially present dental sealants could lower the effectiveness of the program in preventing dental caries development [34]. This low rate could be explained by the age of the children whose first molars were erupting at the time of sealant placement or by the fact that all teeth were sealed and not only teeth at high risk of caries. These rates might also be related to the dentists' lack of involvement in preventive activities. The NC sealant program has been conducted within the French context where the population benefits from Universal Health/Dental Coverage. But dentists are mainly working in private practices with no (or few in NC) community dental services or the presence of associated professionals such as dental hygienists [35]. Historically, priority has been given to operative care, at the expense of preventive dentistry. Many preventive interventions are not covered by the National Health Insurance system. Dentists thus tend to focus on restorative treatments and their management of carious disease is mainly invasive and curatively driven [36].

The dental sealant intervention is integrated within an oral health promotion plan. Evidence of the effect of educational interventions or multi-component school- and community-based interventions is equivocal [3]. Thus, the evaluation of the impact of the program according to the social situation and oral health needs is very important as public health interventions may increase inequalities in the population. The 'inverse care law' states that those most in need of benefiting from preventive interventions are the least likely to receive and benefit from them. Some interventions are successful at improving health across the population but they may increase health inequalities. This can happen when an intervention is of greater benefit to advantaged (lower-risk) groups than to disadvantaged (higher risk) groups [37,38]. Conversely, some interventions may reduce inequalities, if they are of greater benefit to disadvantaged groups. In this study, it was demonstrated that the participation rate was very high whatever the region and that the setting of fissure sealant placement (school vs public dental offices) did not influence the retention rates. The good balance of participation and quality of intervention has been obtained by focusing efforts on ensuring accessibility to the intervention in every place, even the more deprived and isolated ones. In the Islands, caries prevalence was high in 2016 and caries increment was also marked between 2016 and 2017. The mediation analysis showed that having a high retention rate was particularly important for the children in the Islands. Hence, the retention rate was a good intermediate indicator of the effect of the preventive intervention on oral health inequalities, showing that accessible and high-quality interventions are crucial for deprived populations. This is an illustration of the need for proportionate universalism in oral health promotion programs with reinforced resources adapted to the needs of the populations in deprived sectors.

These findings were obtained within a short follow-up period; the calculation of a two-year (or more) caries increment might have led to the measurements of higher impacts. Moreover, it was not possible to make a comparison with a control group as all 6-yr-old children are concerned by the program in NC. Since the Ottawa declaration [8], there has been an expansion

of health promotion programs but there is still a need for an evidence-based evaluation of their impact. Randomized Clinical Trials are needed but also other types of studies such as pragmatic studies conducted in real life contexts. Evaluation studies have to be context sensitive to identify what works for whom and to take into account the implementation processes, and the barriers and facilitators to health promotion programs [39]. Health promotion programs are frequently conducted in school environments or in available community settings. Understanding local contexts as socially-complex systems may help adopt better suited approaches for the dissemination and evaluation of oral health promotion programs [40]. In the present study, sealant application procedures and evaluation process were adapted to the local situation. This has limited the quality of the program and evaluation protocol but, at the same time, this study provided invaluable context-sensitive data that helped understand what works and for whom within a pragmatic perspective.

The oral health promotion program (OHP) conducted in NC since 2014 was developed after the identification of oral health needs, but also after their recognition by the population and major local stakeholders [41]. The sealant application program that is part of the OHP program has been put in place within the context of routine practice. Existing intervention practices were identified and improvements were facilitated. The sealant intervention has been supported first because it is an evidence-based intervention with a solid rationale. Moreover, a user-centred approach was privileged and an interdisciplinary team of researchers, political stakeholders together with health providers were involved in the program. The sealant program was thus integrated in the OHP program as the likelihood of adoption and implementation in daily practice was high.

The sealant program was associated with other interventions aimed at modifying other health determinants such as developing a health education program or promoting tooth brushing in nurseries (age 3–6 years) and schools [42]. These interventions will be evaluated progressively as soon as their implementation has been completed [43]. It must be stated that the level of implementation of tooth brushing at school is very promising (85%) with a high involvement of schools in the deprived areas such as the North region where community workers and teachers became highly involved (https://www.ass.nc). An epidemiological survey was conducted in 2019, among a national representative sample of 6-, 9- and 12-yr-old children. This survey will bring more information about the evolution of children's oral health in NC over the last 5 years and the effect of the OHP program.

## Conclusions

This retrospective study of the quality and impact of the fissure sealant program is part of the evaluation process of a larger OHP program. This study showed a high participation rate and acceptable effectiveness as measured with the one-year retention rates, for a fissure sealant intervention conducted in real-life conditions and integrated in a large health promotion program. Nevertheless, even with well-balanced participation and retention rates between regions, the intervention was not effective enough to totally balance the influence of health determinants, especially in socially-deprived sectors characterized by greater dental needs. These results pointed out the issues of generalizing well-proven effective preventive procedures within specific real-life contexts. The results emphasised the need for continuous and targeted training programs for the program actors to help optimize effectiveness. It would have been illusory to think it possible to counterbalance the effects of social determinants on health with one short-term preventive intervention, even if integrated in a larger health promotion program. Nevertheless, the results show that taking into account the social gradient of health when evaluating public health interventions is a major issue. Further studies to evaluate long-

term participation, quality and effects of the whole health promotion and education program in NC are needed, as well as cost-effectiveness studies to help support sound future public health decisions.

## Supporting information

**S1 Data.**
(XLSX)

## Acknowledgments

The authors would like to thank Caroline Eschevins (CROC, Dental School, Clermont-Ferrand, France) for her help in the writing of the manuscript.

## Author Contributions

**Conceptualization:** Hélène Pichot, Martine Hennequin, Stéphanie Tubert-Jeannin.

**Data curation:** Hélène Pichot, Bruno Pereira, Elodie Magnat.

**Formal analysis:** Hélène Pichot, Bruno Pereira, Elodie Magnat, Stéphanie Tubert-Jeannin.

**Funding acquisition:** Martine Hennequin.

**Investigation:** Hélène Pichot.

**Methodology:** Bruno Pereira.

**Project administration:** Hélène Pichot.

**Writing – original draft:** Hélène Pichot, Bruno Pereira, Elodie Magnat, Stéphanie Tubert-Jeannin.

**Writing – review & editing:** Hélène Pichot, Martine Hennequin, Stéphanie Tubert-Jeannin.

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
