## [Decision Letter · Decision Letter 0]

6 Feb 2020

PONE-D-19-35772

Implementation and impact of a dental preventive intervention conducted within a health promotion program on health inequalities: A retrospective study

PLOS ONE

Dear Pr tubert-jeannin,

Thank you for submitting your manuscript to PLOS ONE. After careful consideration, we feel that it has merit but does not fully meet PLOS ONE’s publication criteria as it currently stands. Therefore, we invite you to submit a revised version of the manuscript that addresses the points raised during the review process.

We would appreciate receiving your revised manuscript by Mar 22 2020 11:59PM. To enhance the reproducibility of your results, we recommend that if applicable you deposit your laboratory protocols in protocols.io, where a protocol can be assigned its own identifier (DOI) such that it can be cited independently in the future. For instructions see: http://journals.plos.org/plosone/s/submission-guidelines#loc-laboratory-protocols

We look forward to receiving your revised manuscript.

Kind regards,

Frédéric Denis, Ph.D.

Academic Editor

PLOS ONE

Journal Requirements:

"The study is an epidemiological retrospective study with a clinical examination of the children at school. The clinical examination was conducted by calibrated dentists and was limited to a visual observation.

Ethical approvals were obtained from the local educational and health institutions. Schools were approached through local educational authorities. Data were recorded in a file registered within the ethical comity for data management (CNIL N° kpP1390145R). Explanatory letters and consent forms were sent to parents prior to the dental examinations and children whose parents returned written consent were examined."

Reviewers' comments:

Reviewer's Responses to Questions

**Comments to the Author**

1. Is the manuscript technically sound, and do the data support the conclusions?

Reviewer #1: Yes

Reviewer #2: Yes

2. Has the statistical analysis been performed appropriately and rigorously? 

Reviewer #1: Yes

Reviewer #2: Yes

3. Have the authors made all data underlying the findings in their manuscript fully available?

Reviewer #1: Yes

Reviewer #2: Yes

4. Is the manuscript presented in an intelligible fashion and written in standard English?

Reviewer #1: Yes

Reviewer #2: Yes

5. Review Comments to the Author

Reviewer #1: a) One major suggestion is that more input & discussion should be developed to contextualize the sentence

327/328 "the intervention was not effective enough to balance the influence of health determinants in socially deprived sectors characterized by high dental needs"

For example analysis should be more deep and address issues like: what are the alternatives that can be introduced to overcome this situation?

b) it would be useful to implement a revision & proof reading of the text since there are some inconsistencies like

line 55 A formation and a standardized - training instead of formation

line 138 Poisson model a used. - Poisson model was used.

line 151 f the children selected - something is missing

lines 227, 229, 230, 232, 234 - etc "dots" misplaced

line 241 punctuation misplaced

line 280 review syntax and grammar

Reviewer #2: Introduction l.6-7 : The authors mention the challenge : taking into acount oral health inequalities gradient when assessing public health interventions. It is very interesting and original to assess the impact of interventions on health inequalities additionaly to a usual global assessment.

P5 l.58 and followings : this challenge does not appear very clearly in the aim of the study. « The aim was also to appreciate the influence of the quality of the intervention (retention rates) on the one-year caries increment, while taking into account initial dental status, conditions of sealants placement and social determinants such as the region of origin. Indeed, in order to tackle health inequalities, a greater or at least equivalent quality of intervention is expected in socially deprived sectors with highest health needs compared to more favoured areas [3]. »

The number of subjects was properly calculated.

P6 l.78 : By « region size », do you mean « region population » ? Did you make a clusterred sample ?

The statistical analysis has a high quality level (normal distribution checking, consequent use of a 0-inflated Poisson model, interactions testings…) and thus is very trustable.

P8 l. 138 : change « thus 0-inflated Poisson model a used » to « thus a 0-inflated Poisson model was used »

P9 l151 : « f the children selected in the 2017 sample participated ». Something might be missing ?

P9 l54 : « Eighty-three percent (95%CI: 79.9%-86.2%) of the dental sealants were present after one year. Among them, 31.1% (27.8%-34.5%) were totally present and 51.9% (48.6%-55.1%) were partially present. » I would suggest to remove « among them ». You don’t mean 31.1% of 83% ? Right ?

P9 l 159 : « Dental status in 2016, as well as the number of sealants applied in 2016, did

not influence retention rates. » This protocol does not allow to proove causal relationships (« influence »). I’d advise to rather talk about « links » or « associations ».

p9 Fig 1 : Why is the total retention rate lower than the complete retention rate ??? There might be a mistake.

Additionnally, it would maybe be clearer to talk about « full retention rate » and « global retention » rate…

p10 Table 1 :

Iles -> Islands

Partial retention rate : 0.52. Wouldn’t it be better to say 52% ? Do you mean 52% of the children had their sealants partially removed ?

What is the statistical unit ? Or do you mean 52% of the sealants ?

If my unsdersanding is right, I’d suggest the following changes :

P7 l 109 : « For each child, the number of sealants applied in 2016 was compared with the number of sealants present during the 2017 examination. This allowed the calculation of the retention rates; total (totally or partially present), partial (partially present) and complete retention rate (totally present). »

« For each child, the numbers of partially remaining sealants and full sealants were recorded. The number of sealants applied in 2016 was compared with the numbers of sealants present during the 2017 examination. This allowed the calculation of 3 retention rates for each child; the complete retention rate (based on fully present sealants), the partial retention rate (partially present) and the global retention rate (fully or partially present). »

If my understanding is right, this sentence seems false to me : as the 2016 number of sealants per child was not always the same, the global percentage is not equal to the mean rate.

P9 l54 « Eighty-three percent (95%CI: 79.9%-86.2%) of the dental sealants were present after one year. Among them, 31.1% (27.8%-34.5%) were totally present and 51.9% (48.6%-55.1%) were partially present. »

I would suggest to change this sentence to : « On average, the children had still 83% (95%CI: 79.9%-86.2%) of their dental sealants after one year. Thirty one point one percent (27.8%-34.5%) of their initial sealants were still totally present and 51.9% (48.6%-55.1%) were partially present. »

If you calculate the global percentage, you get 85%,32% and 53%.

P12 Table 2 : The last column is unreadable : numbers are supeimposed.

P13 : What does all the text lines 224-235 stand for ?

Fig 2 : The definition is low and thus the figure is hard to read. Please provide a more readable figure.

P14 l241 : « and that 80% of the dental sealants were present after one year. » Change to « and that on average, the chidren had 80% of their dental sealants present after one year »

P14 l 252 You calculated the mean retention rate per child and not the global retention rate. The global retention rate woylb even be lower. The studies you are comparing to did calculate the global retention rate.

Your method allows an equal weight of each child in your study.

P15 l 282 : « Hence, it was possible in our sample to show that a high total number of sealants made a decisive contribution to caries prevention, particularly in deprived areas. »

On which result do you ground this assertion ? I do not see a statically significant difference about the « number of sealants » in any of your tables or figures. Indeed (Fig 1) the number of sealants applied inislands is lower than un other regions but the difference is not statically significant.

P17 l 327 « Nevertheless, even with well-balanced participation and retention rates between

regions, the intervention was not effective enough to balance the influence of health determinants in socially deprived sectors characterized by high dental needs. »

« Nevertheless, even with well-balanced participation and retention rates between

regions, the intervention was not effective enough to totally balance the influence of health determinants and especially in socially deprived sectors characterized by high dental needs. »

My conclusion :

This study showed that such a global program neither reaches to totally counterbalance other factors (caries increment 0.12-0.51) nor to alleviate the oral health inequalities (caries increment significantly higher in islands).

Additionnally it showed that independant predicting factors of tooth decay increment seem to be region and full sealant retention. Acting on these two factors might enhance the efficiency of such a program.

Cost-effectiveness studies would be needed to support public health decisions.

This study is a valuable contribution in public health as it is assessing a global program in real life condition.

6. PLOS authors have the option to publish the peer review history of their article (what does this mean?). If published, this will include your full peer review and any attached files.

Reviewer #1: No

Reviewer #2: Yes: Pr Valerie BERTAUD

---

## [Author Response · Author response to Decision Letter 0]

3 Mar 2020

Answer to Editorial comments

Ethics statement

The full name of the “CNIL” comity has been added in the manuscript text and in the ethics statement

Data Availability

The file is available from the Zenodo database (DOI : 10.5281/zenodo.3670886)

English 

Editing revision done, from an official translation office 

Answer to Reviews comments

Reviewer 1

discussion

More text was added in the discussion, to better contextualize and discuss the sentence “”not effective enough to balance the influence of health determinants”

Revision and Proof reading

The corrections requested from L 55 to L280 has been done

Reviewer 2

Objectives

The objectives of the study have been rewritten to better positioned them within the goal of taking into account OH inequalities when assessing public health interventions

Comments

L138 “region population” and “clustered sampling method” added

L138 “was” added.

L151 New wording : “Of the children selected in 2017, 459 children participated”…

L154 “among them” deleted

L159 : “were associated” replaces “influence”

FiG 1 : “Full/complete” instead of “Total” and “Global” instead of “complete” are now used in the whole document . The figure has also been simplified. 

P10 table 1 : 

The word Islands is now used

Retention rates : a sentence has been added to the legend “proportion of sealants remaining per child, “

P7 / L109 The sentence proposed by the reviewer has been integrated

P9 / 152 The sentence proposed by the reviewer has been integrated

P12 Table 2 : The table has been repositioned in the word document

Fig 2 : the quality of the figure has been improved

L241 the proposed change has been made ( idem in the abstract)

L252 the reviewer’s comments have been taken into account in the sentence 

L282 the sentence was grounded on the mediation analysis. Nevertheless, It has been rewritten to better balance its meaning. 

L327 the proposed change has been made

Conclusion : 

The need for cost effectiveness studies has been added to the conclusion

---

## [Editor Report · Decision Letter 1]

5 Mar 2020

Implementation and impact of a dental preventive intervention conducted within a health promotion program on health inequalities: A retrospective study

PONE-D-19-35772R1

Dear Dr. tubert-jeannin,

We are pleased to inform you that your manuscript has been judged scientifically suitable for publication and will be formally accepted for publication once it complies with all outstanding technical requirements.

With kind regards,

Frédéric Denis, Ph.D.

Academic Editor

PLOS ONE
---

## [Editor Report · Acceptance letter]

10 Mar 2020

PONE-D-19-35772R1 

Implementation and impact of a dental preventive intervention conducted within a health promotion program on health inequalities: A retrospective study 

Dear Dr. tubert-jeannin:

I am pleased to inform you that your manuscript has been deemed suitable for publication in PLOS ONE. Congratulations! Your manuscript is now with our production department. 

With kind regards,

on behalf of

Dr. Frédéric Denis 

Academic Editor

PLOS ONE